# Cannabigerol Induces Autophagic Cell Death by Inhibiting EGFR-RAS Pathways in Human Pancreatic Ductal Adenocarcinoma Cell Lines

**DOI:** 10.3390/ijms25042001

**Published:** 2024-02-07

**Authors:** Laura Zeppa, Cristina Aguzzi, Maria Beatrice Morelli, Oliviero Marinelli, Martina Giangrossi, Margherita Luongo, Consuelo Amantini, Giorgio Santoni, Massimo Nabissi

**Affiliations:** 1School of Pharmacy, University of Camerino, Via Madonna delle Carceri 9, 62032 Camerino, MC, Italy; laura.zeppa@unicam.it (L.Z.); cristina.aguzzi@unicam.it (C.A.); mariabeatrice.morelli@unicam.it (M.B.M.); oliviero.marinelli@unicam.it (O.M.); martina.giangrossi@studenti.unicam.it (M.G.); giorgio.santoni@unicam.it (G.S.); 2Integrative Therapy Discovery Lab, University of Camerino, 62032 Camerino, MC, Italy; 3“Maria Guarino” Foundation—AMOR No Profit Association, 80078 Pozzuoli, NA, Italy; 4School of Bioscience and Veterinary Medicine, University of Camerino, 62032 Camerino, MC, Italy; consuelo.amantini@unicam.it

**Keywords:** cannabigerol, pancreatic cancer, autophagy, chemo-resistance, RAS pathways

## Abstract

Pancreatic ductal adenocarcinoma (PDAC) is the most frequent infiltrating type of pancreatic cancer. The poor prognosis associated with this cancer is due to the absence of specific biomarkers, aggressiveness, and treatment resistance. PDAC is a deadly malignancy bearing distinct genetic alterations, the most common being those that result in cancer-causing versions of the KRAS gene. Cannabigerol (CBG) is a non-psychomimetic cannabinoid with anti-inflammatory properties. Regarding the anticancer effect of CBG, up to now, there is only limited evidence in human cancers. To fill this gap, we investigated the effects of CBG on the PDAC cell lines, PANC-1 and MIAPaCa-2. The effect of CBG activity on cell viability, cell death, and EGFR-RAS-associated signaling was investigated. Moreover, the potential synergistic effect of CBG in combination with gemcitabine (GEM) and paclitaxel (PTX) was investigated. MTT was applied to investigate the effect of CBG on PDAC cell line viabilities. Annexin-V and Acridine orange staining, followed by cytofluorimetric analysis and Western blotting, were used to evaluate CBG’s effect on cell death. The modulation of EGFR-RAS-associated pathways was determined by Western blot analysis and a Milliplex multiplex assay. Moreover, by employing the MTT data and SynergyFinder Plus software analysis, the effect of the combination of CBG and chemotherapeutic drugs was determined.

## 1. Introduction

Pancreatic ductal adenocarcinoma (PDAC) is an infiltrating neoplasm derived from the pancreatic ductal tree, and its 5 year overall survival rate increased from 3% to 9% in the last 50 years [1]. The worst prognosis is related to the absence of specific symptoms, diagnosis biomarkers, high invasiveness, and treatment resistance [2]. The PI3K/AKT/mTOR and RAF/MEK/ERK pathways, frequently activated in PDAC, are involved in mediating autophagy, apoptosis, and chemoresistance, and they are known to drive PDAC development and tumor maintenance [3]. While pharmacological approaches to block RAS signaling were clinically unsuccessful, as demonstrated by no clinical benefit of monotherapy in trials [4], efforts are focusing on downstream proteins, such as RAF, MEK, and Akt [5]. Different PI3K/Akt/mTOR inhibitors are under clinical evaluation, but their antitumoral effects may be limited by the over-activation of RAS/RAF/MEK signaling [6]. A novel strategy to combine inhibitors of these pathways could be more effective, and some preclinical studies have evidenced improved efficacy in various tumor types [7]. Regarding PDAC, this adjuvant therapy was demonstrated to improve overall survival but in a limited number of patients as combination chemotherapy regimens (mFOLFIRINOX or gemcitabine/capecitabine) have been shown to be more effective than single-agent chemotherapy in the adjuvant treatment of PDAC [8]; thus, new pharmacological and combination therapies are needed. Phytocannabinoids, such as ∆9-Tetrahydrocannabinol (THC) and Cannabidiol (CBD), compounds derived from *Cannabis sativa* L., as well as medical cannabis, were evidenced to exert antitumoral activity when used in combination with other therapies in glioblastoma, as reported in one clinical trial and in one case report [9,10]. Moreover, many in vitro and in vivo preclinical studies evidenced that phytocannabinoids show antitumoral effects in different human cancers, and synergistic effects are observed with approved chemotherapeutic drugs, as recently reviewed in [11].

The non-psychomimetic phytocannabinoid, cannabigerol (CBG), was evaluated for its potential anticancer effect in human cancer cell lines, including melanoma and glioblastoma [12,13]. Although the pathways activated by CBG were not yet clarified, recently, CBG was demonstrated to bind the kinase active site of EGFR and inhibit its intracellular kinase activity, inducing apoptosis in the EGFR-overexpressing A549 and A431 cell lines [14]. This is particularly important because the EGFR protein family is involved in the initialization and progression of PDAC, and EGFR inhibitors are used in clinical trials as anticancer therapy [15].

In this study, CBG’s anticancer effect was investigated in two human PDAC cell lines, PANC-1 and MIAPaCa-2, in order to analyze its interactions with the EGFR-RAS related pathways involved in sustaining cell viability and chemoresistance.

## 2. Results

### 2.1. CBG Inhibits Cell Growth of PDAC Cell Lines

PANC-1 and MIAPaCa-2 PDAC cell lines and immortalized NCM460D and NHFA-12 cell lines were treated with different doses of CBG (up to 31.648 μg/mL) as single administration, and cell viability was determined by the MTT assay at 72 h post-treatment. The results showed that CBG reduces cell viability in both cell lines with an IC_50_ of 15.64 ± 0.83 μg/mL for PANC-1 and 13.77 ± 0.72 μg/mL for MIAPaCa-2 (Figure 1), while lower cytotoxicity was observed in normal human NCM460D and NHFA12 cell lines (Appendix A). For the next experiments, we selected two doses of CBG (11.08 and 12.66 µg/mL), representing lower cytotoxic doses used for studying molecular and biological mechanisms in the PDAC cell lines with no cytotoxic effects on normal cells. CBG was always administered as a single dose.

### 2.2. CBG Induces Autophagy by Inhibition of EGFR and Akt/mTOR Pathway in PDAC Cell Lines

EGFR and Akt/mTOR protein levels were evaluated by Western blot in PDAC cell lines at 48 h post-treatment with CBG doses of 11.08 and 12.66 µg/mL. Immunoblots evidenced a decrease in EGFR expression in PANC-1 with the higher dose of CBG, while, for MIAPaCa-2, the reduction was significant with both doses (Figure 2A). Similar results were obtained for mTOR protein expression, which was reduced especially after the treatment with the highest dose of CBG (Figure 2A). Then, total Akt and its phosphorylated form were investigated. Data showed a slight modulation of total Akt protein with CBG at a dose of 11.06 μg/mL for PANC-1 cells and a reduction of phospho-Akt (pAkt) levels after the administration of both CBG doses (Figure 2B). In MIAPaCa-2 cells, the highest dose of CBG induced a marked reduction of total Akt and a significant decrease in its phosphorylation (Figure 2B).

Since CBG reduced the Akt/mTOR signaling pathway, autophagy induction was also evaluated. We observed the conversion of microtubule-associated protein 1A/1B-light chain 3 (LC3-I) to the LC3-II form at 48 h post-treatment with CBG (11.08 and 12.66 µg/mL). In both cell lines, CBG induced a significant increase in the LC3-II lipidated form, suggesting the activation of the early steps of autophagy; MIAPaCa-2 cells showed the highest levels of LC3-II at the highest dose of CBG (Figure 3A).

To confirm the induction of autophagy, acridine orange dye (AO) analysis, which accumulates in acidic spaces and emits bright red fluorescence related to the degree of the acidity and the volume of acidic vesicular organelles (AVOs), was applied. AVOs are indicative of autophagy; therefore, we evaluated AVOs under a confocal fluorescence microscope. As shown (Figure 3B), an increase in the level of red fluorescent signals reflecting AVO formation was observed in CBG-treated cell lines 48 h post-treatment.

### 2.3. CBG Reduces RAS Downstream Pathway in PDAC Cell Lines

To further investigate the CBG-induced effect in inhibiting the EGFR pathway, the expression of total RAS (Pan-RAS) was assessed by Western blot analysis. As evidenced in Figure 4A, Pan-RAS levels were statistically reduced at 48 h post-CBG treatment. Due to the relevance of the RAS/RAF/MEK pathway in PDAC, we measured the signaling proteins downstream of RAS in the lysate of PDAC cells treated with CBG at the most effective dose of 12.66 μg/mL. By using the MILLIPLEX^®^ RAS-RAF Oncoprotein Magnetic Bead Panel 6-plex, we evaluated changes in phosphorylated BRAF (Ser446, pBRAF), CRAF (Ser338, pCRAF), and MEK1 (Ser217/Ser221, pMEK1), as well as the relative total protein levels (Figure 4B). The results confirmed that CBG treatment reduced total Pan-RAS levels in both cell lines. It also promotes the downregulation of BRAF, CRAF, and MEK1 phosphorylation in PANC-1 and BRAF in the MIAPaCa-2 cell line. In summary, these data further confirm that CBG was able to interfere with the EGFR-RAS pathways by also reducing downstream effectors that are often aberrantly activated in PDAC.

### 2.4. CBG Induces Apoptosis in PDAC Cell Lines

To further investigate whether reduction of cell viability and activation of autophagy was associated with cell death, PANC-1 and MIAPaCa-2 cells were treated with 11.08 and 12.66 µg/mL of CBG, and Annexin V–FITC staining analysis were performed 48 h post-treatment. The results showed that CBG increases Annexin V^+^ cells in a dose-dependent mode (Figure 5A), suggesting the induction of apoptotic cell death. The activation of ProCaspase-3 (ProCasp-3) and the presence of the cleaved formed Caspase-3 (Casp-3) was investigated by Western blot analysis. Blots showed an increase in the Casp-3/ProCasp-3 ratio after 48 h post-treatment with CBG (12.66 µg/mL) in both cell lines (Figure 5B).

### 2.5. The Combination of CBG with PTX or GEM Induces Higher Cytotoxicity Compared to Administration of Single Drugs in PDAC Cell Lines

To support the potential use of CBG as integrative therapy in PDAC, we evaluated the combined effects of CBG with PTX and GEM by MTT assay. Both cell lines were treated with CBG (7.91, 12.66, and 15.82 µg/mL) in combination with three cytotoxic doses of PTX (1.5, 3, and 6 µg/mL) or GEM (25, 50, and 100 µg/mL) and cell viability was evaluated by MTT assay at 72 h post-treatment. The results showed that the combinations with the two higher doses of CBG resulted in greater cytotoxicity when compared to PTX or GEM alone in both cell lines (Figure 6A,B). Then, the drug interaction was evaluated with SynergyFinder using the Bliss model. We obtained a Bliss synergy score of 4.40 for PTX and CBG and 10.87 for GEM and CBG in the PANC-1 cell line, indicating additive and synergistic effects, respectively. In the MIAPaCa-2 cell line, the Bliss synergy score was 17.19 for PTX and CBG and 14.05 for the GEM and CBG combination, suggesting a synergistic effect (Figure 6C,D).

## 3. Discussion

PDAC’s poor prognosis is mainly due to an aggressive behaviour associated with chemoresistance. The search for integrative therapies that can ameliorate chemotherapy’s side effects or improve their effect is constantly under study [16]. *Cannabis sativa* L. contains more than 100 phytocannabinoids and, for some of them, several biological properties are well known. Besides a direct anticancer effect, mainly demonstrated for CBD, in in vitro and in vivo experiments, phytocannabinoids were suggested to ameliorate numerous important side effects induced by chemotherapeutics [17]. In this study, CBG showed dose-dependent cytotoxicity in PDAC cells, as also evidenced in other human preclinical cancer models, such as in glioblastoma multiforme (GBM) [13]. Further, THC, CBD, and synthetic cannabinoids reduced PANC-1 and MIAPaCa-2 cell growth and viability, as reported in several studies and, in line with our evidence, MIAPaCa-2 was more sensitive than PANC-1 to treatments with phytocannabinoids [18,19]. In several pathologies, including cancer, such as seen in glioma cells, cannabinoids have been demonstrated to activate autophagy and apoptotic cell death through the interaction between apoptosis and autophagy signaling mechanisms [20,21]. Several pathways are mediated through the multiprotein complex involved in EGF/EGFR, including RAS and mTOR. These pathways suppress autophagy and promote proliferation and resistance to chemotherapy [22]. The simultaneous inhibition of EGFR and RAS/mTOR was demonstrated to provide a synergistic antitumor effect in various human cancers. Indeed, the PI3K/AKT/mTOR axis, a frequently dysregulated pathway in PDAC, is responsible for the control of cell proliferation and resistance [23], and these pathways can be inhibited by cannabinoids [20,24]. The present data evidenced that CBG induced autophagy by reducing the Akt-mTOR pathways with consequent LC-3 conversion and autophagic vesicle formation, as previously observed with other phytocannabinoids. For example, THC inhibits AKT/mTOR [21], reducing the proliferation of glioma cells. A similar finding was observed in hepatocarcinoma cells, where THC inhibits AKT/mTORC1 through ER stress-dependent activation of AMPK [25]. In our results, CBG reduced mTOR protein expression and, in line with this, some studies demonstrated that CBD inhibits the mTOR signaling pathway in breast cancer and human glioma [24,26]. The observed autophagy mechanism can be mediated by the EGFR and Akt/mTOR signaling axis. We also investigated the modulation of pro-autophagic markers and the results showed that CBG increased LC3-II expression mainly in the more sensitive MIAPaCa-2 cells, where the increase was very noticeable. Autophagy has a double function: to induce cancer resistance to chemotherapy and protect cancer cells from death or to be correlated with cancer cell death. Herein, we evidenced that CBG induced autophagy and apoptosis in both cell lines. Cannabinoids are also involved in reducing cancer cell growth by modulating the EGFR-RAS-RAF-MAPK pathway. In pancreatic cancer, mutated *KRAS* upregulates endogenous EGFR expression, and hyperactivation results in a transformation from acinar to ductal metaplasia [27]. Herein, we showed that CBG suppresses EGFR expression in PANC-1 and MIAPaCa-2. Up to now, there are no data indicating the ability of CBG to reduce EGFR expression; however, some studies demonstrated that CBD and THC can reduce EGFR expression in A549, H460, and H1792 cells and suppress EGF/EGFR signaling pathways in breast cancer [28]. Moreover, our data evidenced that CBG was able to reduce downstream RAS signaling, suggesting a specific role in decreasing the RAS oncogenic pathways in PDAC. In our studies, Annexin V positive cells and Caspase-3 cleavage confirmed CBG induction of apoptosis. In line with our results, CBG induced Caspase-3/-7-dependent apoptosis in glioblastoma and Caco-2 cells [13,14]. Moreover, in PANC-1 and MIAPaCa-2 cells, CBD also induced apoptosis and Caspase-3 activation [19]. Lastly, many studies demonstrated that cannabinoids could increase the efficacy of chemotherapeutic drugs, reducing tumor growth and overcoming drugs resistance [11]. Herein, the combination of CBG with GEM or PTX increased the cytotoxicity compared to the administration of the drug alone, also showing a synergistic effect for some combinations. In GBM cells, CBD and CBG plus temozolomide did not show an additive effect, but in cholangiocarcinoma cells, CBG synergized with GEM and cisplatin [13,29]. Moreover, in PDAC, CBD showed the ability to increase GEM and PTX efficacy in in vitro tests and KPC mice treated with CBD and GEM showed a survival three times longer than mice treated with GEM [18,19]. Overall, our data evidenced the ability of CBG to induce autophagy, reduce EGFR/AKT/RAS pathways, promote apoptotic cell death, and increase the sensitivity of PDAC cell lines to chemotherapeutic drugs. Further study on CBG will be necessary to better understand the role of these compounds in PDAC progression, and in vivo study will be useful to investigate its anticancer effects more deeply.

## 4. Materials and Methods

### 4.1. Cell Lines

The PANC-1 and MIAPaCa-2 human pancreatic ductal adenocarcinoma cell lines purchased from Sigma Aldrich (Milan, Italy) were cultured in DMEM high glucose medium (EuroClone, Milan, Italy) supplemented with 10% of fetal bovine serum (FBS), 2 mM L-glutamine, 100 IU/mL penicillin, 100 mg streptomycin, and 1 mM sodium pyruvate. Epithelial cell lines derived from the normal human colon (NCM460D) and normal human fibroblast (NHFA12) were used as normal cells for the cytotoxicity assay. NCM460D was cultured in RPMI1640 supplemented with 10% of fetal bovine serum (FBS), 2 mM L-glutamine, 100 IU/mL penicillin, and 100 mg streptomycin. NHFA12 was cultured for the PANC-1 cell line. Cell lines were maintained at 37 °C with 5% CO_2_ and 95% humidity.

### 4.2. Reagents

Pure CBG (≥98% purity) was purchased (Cayman Chemical, Ellsworth, MI, USA, cat. No. 36975). CBG was solubilized in 70% ethanol (Et-OH) at a concentration of 50 mM (15.8 mg/mL). Paclitaxel (PTX, 6 mg/mL; cat. No. 33069-62-4) and Gemcitabine (GEM, 50 mg/mL; cat. No. 122111-03-9) supplied by Sigma Aldrich, Milan, Italy were dissolved in water. All the compounds were aliquoted, stored at −20 °C, and each aliquot was used only once.

### 4.3. MTT Assay

In total, 3 × 10^4^ cells/mL were seeded in a 96-well plate. The following day, compounds or vehicles were added, and six replicates were used for each treatment. At 72 h post-treatment, cell viability was analyzed, as previously reported [16]. The absorbance of the sample was measured at 570 nm using an ELISA reader microliter plate (BioTek Instruments, Winooski, VT, USA).

### 4.4. Annexin V Staining

Annexin V-FITC staining and cytofluorimetric analysis were applied for the detection of apoptotic cells in PDAC cell lines. Overall, 3 × 10^4^ cells/mL were treated with different doses of CBG. At 48 h post-treatment, the cells were stained with Annexin V-FITC (5 μL; Adipogen Life Sciences, San Diego, CA, USA, No. AG-40B-0005F), and the percentage of positive cells was analyzed by a FACScan flow cytometer using the CellQuest software (BD Biosciences, Milan, Italy).

### 4.5. Western Blot Analysis

The PDAC cell lines were treated with CBG and harvested at 48 h post-treatment. Lysates were obtained with a lysis buffer (TRIS 1M pH 7.4, NaCl 1M, EGTA 10 mM, NaF 100 mM, Deoxycholate 2%, EDTA 100 mM, TritonX-100 10%, Glycerol, SDS 10%, Na_2_P_2_O_7_ 1 M, Na_3_VO_4_ 100 mM, PMSF 100 mM, cocktail of enzyme inhibitors, and H_2_O), separated on SDS polyacrylamide gel, and Western blots were performed as previously reported [20]. The following antibodies were used: mouse anti-mTOR (1:1000, GeneTex Alton Pkwy Irvine, CA, USA), mouse anti-EGFR (1:5000, GeneTex, Alton Pkwy Irvine, CA, USA), rabbit anti-pAkt (1:1000, Cell Signaling, Danvers, MA, USA), rabbit anti-Akt (1:1000, Cell Signaling, Danvers, MA, USA), rabbit anti-LC3 (1:1000, Cell Signaling, Danvers, MA, USA), rabbit anti-Caspase-3 (1:1000, Cell Signaling, Danvers, MA, USA), mouse pan-RAS (1:500, Santa Cruz Biotechnology, Dallas, TX, USA), mouse β-actin (1:500, Santa Cruz Biotechnology, Dallas, TX, USA), and mouse anti-glyceraldehydes-3-phosphate dehydrogenase (GAPDH, 1:1000, Santa Cruz Biotechnology, Dallas, TX, USA). The Abs were incubated overnight or for 1 h according to the manufacturer’s protocol and then incubated with their respective HRP-conjugated anti-rabbit or anti-mouse (1:2000, Cell Signaling, Danvers, MA, USA) Abs for 1 h. Peroxidase activity was visualized with the LiteAblot^®^ PLUS or TURBO (EuroClone, Milan, Italy) kit and densitometric analysis was carried out by a Chemidoc using the Quantity One software version 4.6 (Bio-Rad, Milan, Italy).

### 4.6. Acridine Orange Staining

To detect acidic vesicular organelles, the vital staining of cells with acridine orange (AO, Sigma-Aldrich, Milan, Italy) was performed. In total, 3 × 10^4^ cells/mL were seeded in 12-well plates and treated with CBG at 11.08 and 12.66 µg/mL for 48 h. Then, cells were stained with 1 μg/mL AO, washed in PBS, immobilized on slides using the cytospin centrifuge and analyzed with a C2 Plus confocal laser scanning microscope (Nikon Instruments, Florence, Italy). The optimized emission detection bandwidth was configured by using the Zeiss Zen 2.0 control software. In addition, images were processed using the NIS Element Imaging software (Nikon Instruments, Florence, Italy). The cytoplasm and nuclei of AO-stained cells fluoresced bright green, whereas the acidic autophagic vacuoles fluoresced bright red.

### 4.7. Milliplex Multiplex Assay

The levels of total RAS, pBRAF, pCRAF, and pMEK1 in treated cells were measured using the RAS-RAF Oncoprotein Panel 6-Plex Magnetic Bead Kit 96-well Plate (EMD Millipore Corporation, Billerica, MA, USA) following the manufacturer’s protocol. Data were analyzed using Luminex^®^ 200 instrument with the xPONENT^®^ 4.2 software (Luminex Corporation, Austin, TX, USA).

### 4.8. Statistical Analysis

The data presented represent the mean and standard deviation (SD) of at least 3 independent experiments. The statistical significance was determined by two-way Anova followed by Dunnett’s or Sidak’s multicomparison test using the GraphPad 9.0.1 software.

### 4.9. Drug Interaction

Drug interaction was evaluated with SynergyFinder using the Bliss model [30]. The Bliss independence reference model was used for the multiplicative effect of single drugs, thus testing the effects of these drugs as if they acted independently. A Bliss synergy score larger than 10 is considered synergistic, a score from −10 to 10 is considered additive, and a score less than −10 is considered antagonistic.

## 5. Conclusions

In conclusion, our results showed that CBG, a non-psychomimetic cannabinoid from *Cannabis Sativa* L., can induce an anticancer effect in two human PDAC cell lines, supporting the ability of cannabinoids to interfere with several pro-tumoral pathways. Further study in pre-clinical in vivo models will be performed to better understand CBG’s effects on PDAC progression.

## Figures and Tables

**Figure 1 ijms-25-02001-f001:**
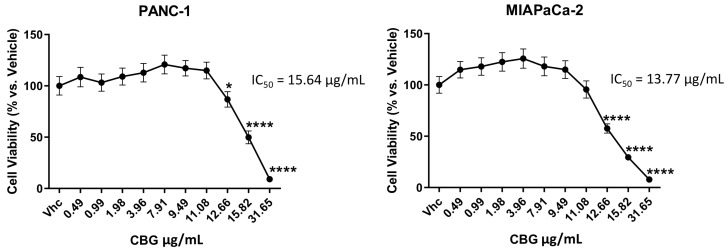
CBG reduces cell viability in PDAC cell lines. Cell viability was determined by MTT assay. PANC-1 and MIAPaCa-2 cells were treated with vehicle (Vhc; Et-OH 70%) or different concentrations of CBG, and cell viability was evaluated at 72 h post-treatment. Data shown are the mean ± SD of three separate experiments. * *p* < 0.05, **** *p* < 0.0001 vs. Vhc.

**Figure 2 ijms-25-02001-f002:**
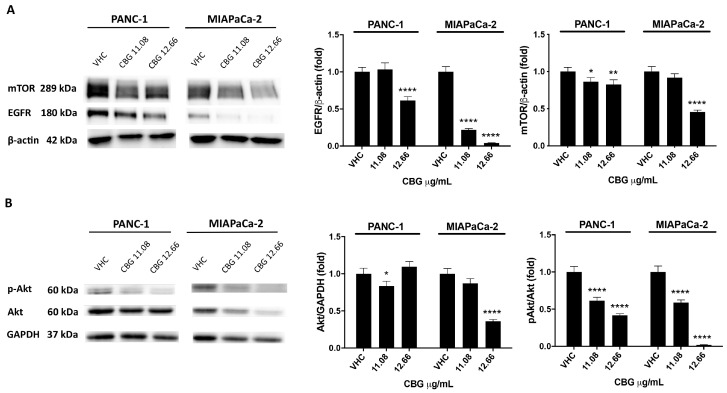
CBG treatment reduces EGFR and Akt/mTOR protein expression in PDAC cell lines. PDAC cell lines were treated with CBG, and the expression levels of mTOR and EGFR (**A**), Akt, and pAkt (**B**) were evaluated by Western blot at 48 h post-treatment. EGFR, Akt, and mTOR densitometric values were normalized to β-actin or glyceraldehyde-3-phosphate dehydrogenase (GAPDH), which were used as loading controls; pAkt densitometric values were normalized to Akt. Images are representative of one of three separate experiments. Data are expressed as mean ± SD of three separate experiments. * *p* < 0.05, ** *p* < 0.01, **** *p* < 0.0001 treated cells vs. VHC.

**Figure 3 ijms-25-02001-f003:**
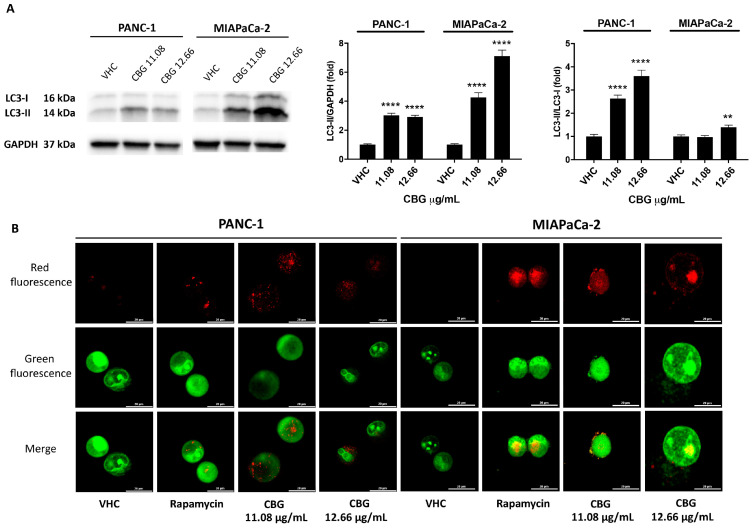
CBG stimulates autophagy in PDAC cell lines. PDAC cell lines were treated with CBG and analyzed at 48 h post-treatment. (**A**) The expression of LC3-I and LC3-II was assessed by Western blot. Densitometric values of LC3-II were normalized to GAPDH, which was used as loading control. Densitometric values of LC3-II were also normalized to LC3-I. Images are representative of one of three separate experiments and data are expressed as the mean ± SD of three separate experiments. ** *p* < 0.01, **** *p* < 0.0001 CBG- vs. VHC-treated cells. (**B**) Fluorescent microscope was used to visualize AVOs (red fluorescence) as well as the cytoplasm and nucleus (green fluorescence) after the vital staining of the cells with AO. Representative images of PANC-1 and MIAPaCa-2 cells stained with AO are shown. Rapamycin was used as positive control of autophagic induction.

**Figure 4 ijms-25-02001-f004:**
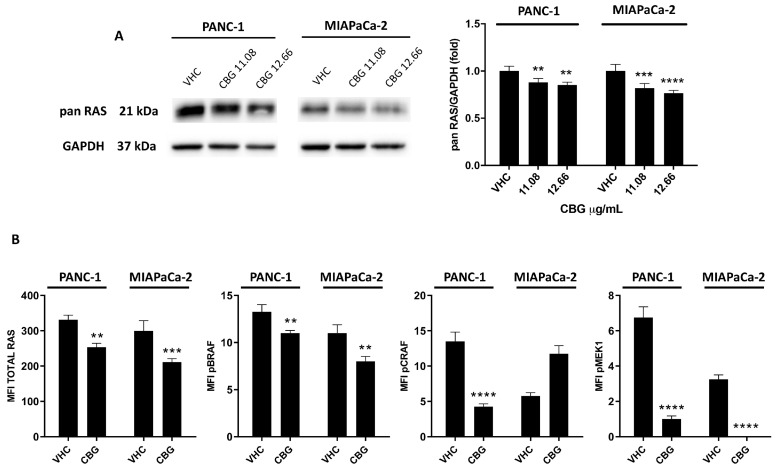
CBG reduces the total Pan-RAS expression and the expression of phosphorylated forms of BRAF, CRAF, and MEK1 in PDAC cell lines. (**A**) PDAC cell lines were treated with CBG, and the expression of Pan-RAS was determined with Western blot at 48 h post-treatment. The densitometric values were normalized to GAPDH used as loading control. Images are representative of one of three separate experiments and data are expressed as the mean ± SD of three separate experiments. ** *p* < 0.01, *** *p* < 0.001, **** *p* < 0.0001 treated cells vs. VHC. (**B**) PDAC cell lines were treated with 12.66 µg/mL CBG for 48 h and the expression of RAS, pBRAF, pCRAF, and pMEK1 was evaluated by Milliplex multiplex assay, and the median fluorescence intensity (MFI) was measured with the Luminex^®^ system. Data shown are expressed as the mean ± SD of three separate experiments. ** *p* < 0.01, *** *p* < 0.001, **** *p* < 0.0001 treated cells vs. VHC.

**Figure 5 ijms-25-02001-f005:**
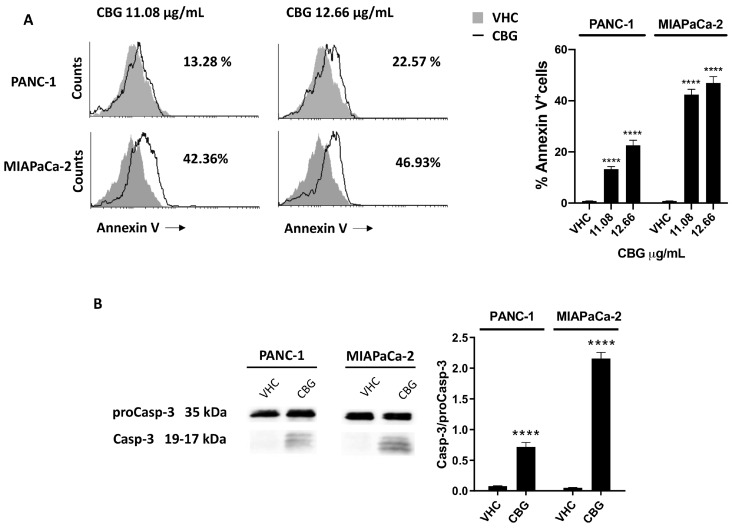
CBG-induced apoptotic cell death in PDAC cell lines. PDAC cell lines were treated with CBG and analyzed at 48 h post-treatment. (**A**) Flow cytometric analysis was performed after Annexin V staining. Images are representative of one of three separate experiments and histograms are the mean ± SD of three separate experiments. The percentage refers to Annexin V^+^ cells observed in CBG- vs. VHC-treated cells. (**B**) The expression of Caspase-3 was determined by Western blot. Caspase-3 densitometric values were normalized to proCaspase-3. Image is representative of one of three separate experiments and bars represent the mean ± SD of three separate experiments. **** *p* < 0.0001 treated cells vs. VHC.

**Figure 6 ijms-25-02001-f006:**
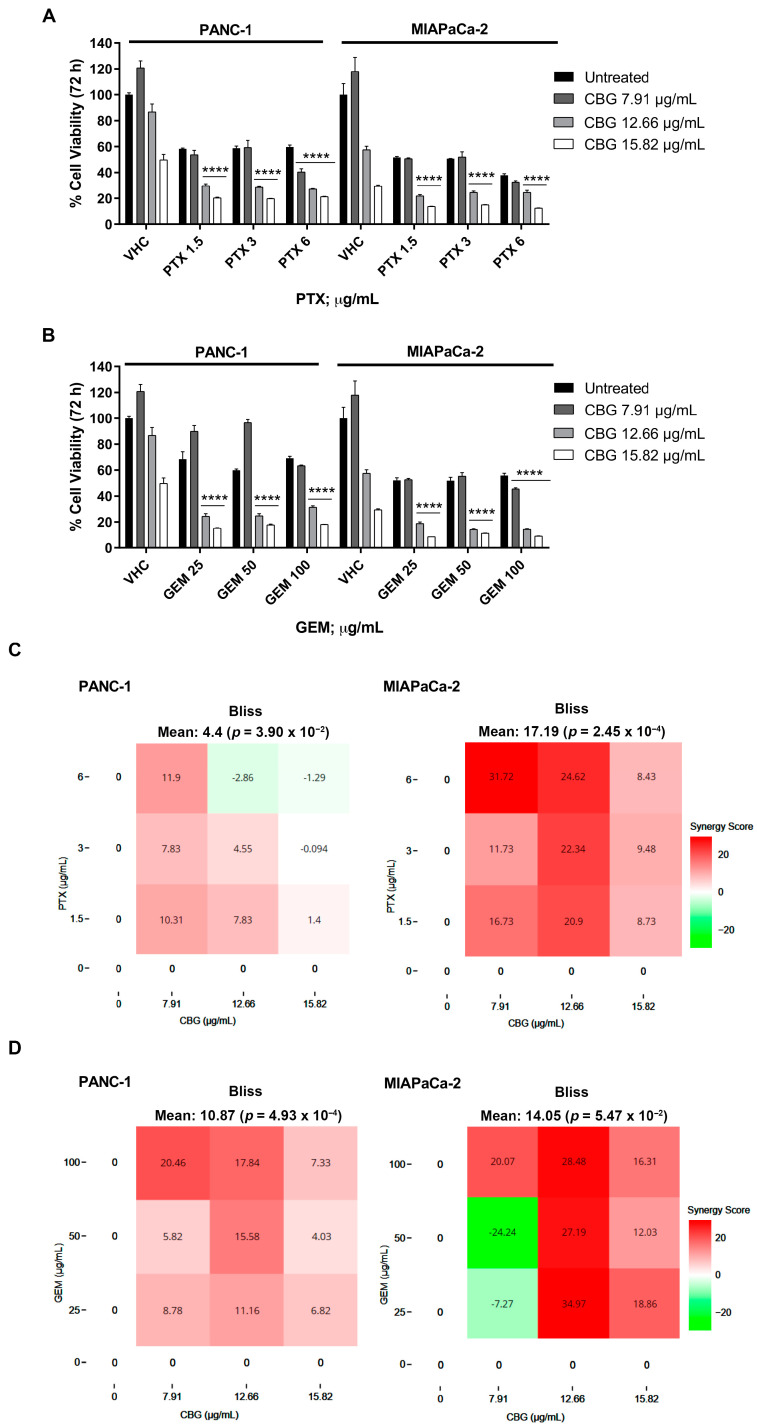
CBG increased the cytotoxic effect of chemotherapeutic drugs in PDAC cell lines. (**A**,**B**) Cell viability was determined in PDAC cell lines by MTT assay. Cells were treated for 72 h with CBG alone or in combination with different doses of GEM and PTX. Data shown are expressed as mean ± SD of three separate experiments. **** *p* < 0.0001 CBG combined with chemotherapeutic drug vs. chemotherapeutic drug alone. (**C**,**D**) Drug interaction of CBG with PTX and GEM in PDAC cell lines. Effect of single and combined treatments with CBG (7.91, 12.66, 15.82 µg/mL), PTX (1.5, 3, 6 µg/mL), and GEM (25, 50, 100 µg/mL) on cell viability of PDAC cell lines. Drug interaction was evaluated with SynergyFinder software using the Bliss model. Bliss synergy score larger than 10 is considered synergistic, a score from −10 to 10 is considered additive, and a score less than −10 is considered antagonistic.

## Data Availability

The data presented in this study are available on request from the corresponding author.

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
