# Peer review of "Cannabigerol Induces Autophagic Cell Death by Inhibiting EGFR-RAS Pathways in Human Pancreatic Ductal Adenocarcinoma Cell Lines"

_ijms, 2024, doi:10.3390/ijms25042001_

Round 1

Reviewer 1 Report

Comments and Suggestions for Authors

The paper on Cannabigerol presented to me for review is well planned and well written. Basically, I have no substantive comments on it, only a few editorial comments.

Figures in the work are difficult to read. Their size should either be increased, or at least the font in the drawings should be enlarged. The volume unit mL should be written the same throughout the work (sometimes it is mL, sometimes ml). Citations should also be edited in accordance with the journal's guidelines. They are missing doi numbers.

In conclusion, I believe that the work requires only minor corrections and can be published.

Author Response

Dear reviewer,

thanks for the comments. We have improved the figures quality. The errors were corrected as requested. The DOI has been added to each reference.

Reviewer 2 Report

Comments and Suggestions for Authors

Dear colleagues,

The manuscript “Cannabigerol induces autophagic-cell death by inhibiting EGFR-RAS-pathways in human pancreatic ductal adenocarcinoma cell lines” by Zeppa L. et al is a research article that showed interesting preclinical results for pancreatic ductal adenocarcinoma (PDAC) treatment, by investigating the role of cannabigerol (CBG) and its effects on PDAC cell lines (PANC-1 and MIAPaCa-2). Specifically, the authors conducted a series of experiments in order to evaluate and quantify the potential anticancer effects of CGB on EGFR/RAS pathways typically involved in PDAC development and growth. Also, the potential synergistic antitumor effect of CBG in combination with gemcitabine and paclitaxel was investigated in aforementioned cell lines.

While the manuscript itself appears to be interesting, it has some issues that should be addressed:

1.       English requires minor revision.

2.       The order of the paragraphs seems to be misleading. I suggest to discuss the materials and methods right after the introduction, in order to support interpretation of the results.

3.       All the figures (1-6) seem not to be in their final version and appear doubled. Please correct.

4.       Many times throughout the manuscript and figures’ descriptions the authors refer to values expressed as means and standard deviations. However, standard deviations are not reported neither in the text nor in the graphics.

5.       In line 46 an adjuvant therapy is mentioned. Please give further details.

6.       In paragraph 2.1, it is said that found IC50 was 15,64 µg/ml for PANC-1 and 13,77 µg/ml for MIAPaCa-2. However, the next sentence stated that 11,08 µg/ml and 26,66 µg/ml were the doses selected for further experiments. Please explain the process behind this choice.

7.       In paragraph 2.2, it is said that “Data showed for PANC-1 cells a slight modulation of total Akt 95 protein and a reduction of phospho-Akt levels after the administration of CBG” (line 95). Please explain at which dose of CBG this happened. Moreover, it would be helpful to cite the part of the figure (A or B) associated with the text. Also, I suggest to mention results in the same order as they appear in the graphics.

8.       In Figure 5, it is said that “PDAC cell lines were treated with CBG for 48h”, please explain.

9.       In paragraph 2.5, it is said that “Both cell lines were treated with CBG 7,91, 12,66 and 15,82 μg/mL” (line 198). Please explain why these three doses were chosen.

10.   Please correct bibliography in line 318.

11.   Finally, I suggest to expand the discussion, also focusing on the limitations of the results and the implications of the findings for further research in oncological field.

Kind regards

Comments on the Quality of English Language

I suggest minor revision.

Author Response

English requires minor revision.

- Thanks, we revised the text.

The order of the paragraphs seems to be misleading. I suggest to discuss the materials and methods right after the introduction, in order to support interpretation of the results.

- The MS was organized according to the instruction for authors: Research manuscript sections: Introduction, Results, Discussion, Materials and Methods, Conclusions (optional).

All the figures (1-6) seem not to be in their final version and appear doubled. Please correct.

- The final versions were added to the text.

Many times throughout the manuscript and figures’ descriptions the authors refer to values expressed as means and standard deviations. However, standard deviations are not reported neither in the text nor in the graphics.

- The standard deviation was added in Result (section 2.1), were IC50 was described. The Stand. Dev. of the experiments were represented by the bars in the histograms.

In line 46 an adjuvant therapy is mentioned. Please give further details.

- Thanks, we added more info in the text.

In paragraph 2.1, it is said that found IC50 was 15,64 µg/ml for PANC-1 and 13,77 µg/ml for MIAPaCa-2. However, the next sentence stated that 11,08 µg/ml and 26,66 µg/ml were the doses selected for further experiments. Please explain the process behind this choice.

- The doses 11,08 and 12,66 (sub-IC50) were selected since to study the CBG effects at molecular levels, was necessary to induces the process (ex. Cell death, autophagy) but working with live cells.

In paragraph 2.2, it is said that “Data showed for PANC-1 cells a slight modulation of total Akt 95 protein and a reduction of phospho-Akt levels after the administration of CBG” (line 95). Please explain at which dose of CBG this happened. Moreover, it would be helpful to cite the part of the figure (A or B) associated with the text. Also, I suggest to mention results in the same order as they appear in the graphics.

- Thanks, the doses have been added and the part of the figures have been cited in the text. The result from Figure 2 were modified as requested.

In Figure 5, it is said that “PDAC cell lines were treated with CBG for 48h”, please explain. 

- Thanks, we correct the sentence with 48 h post-treatment.

In paragraph 2.5, it is said that “Both cell lines were treated with CBG 7,91, 12,66 and 15,82 μg/mL” (line 198). Please explain why these three doses were chosen.

- For synergism analysis, the software required three doses for each compound. So, we selected a non-cytotoxic dose (7,91 μg/mL), and two doses with cytotoxicity, according to MTT data.

Please correct bibliography in line 318.

- Apologize, we don’t find which reference should be corrected.

Finally, I suggest to expand the discussion, also focusing on the limitations of the results and the implications of the findings for further research in oncological field.

- Thanks, we improved the discussion focusing on the limitation of this work and implication for further reseach.

Reviewer 3 Report

Comments and Suggestions for Authors

Very good manuscript demonstrating the apoptotic and cytotoxic effect of Cannabigerol on 2 pancreatic carcinoma cell lines and the molecular mechanism involved. The authors found also the concentration to be used, a very important issue while working with cannabinoids. The dramatic inhibitory effect on pancreatic carcinoma is impressive.  In this study, CBG anticancer effect was investigated in two human PDAC cell lines,

62
PANC-1 and MIAPaCa-2, in order to analyse its interactions with the EGFR-RAS related 63
pathways involved in sustaining cell viability and chemoresistance.

Author Response

Dear reviewer,

thanks for the comments.